# Validation of chronic obstructive pulmonary disease recording in the Clinical Practice Research Datalink (CPRD-GOLD)

Jennifer K Quint,[1] Hana Müllerova,[2] Rachael L DiSantostefano,[3] Harriet Forbes,[1] Susan Eaton,[4] John R Hurst,[5] Kourtney Davis,[2] Liam Smeeth[1]

▶ Prepublication history and additional material is available. To view please visit the journal (http://dx.doi.org/10.1136/bmjopen-2014-005540).

For numbered affiliations see end of article.

**Correspondence to**
Dr Jennifer K Quint;
Jennifer.quint@lshtm.ac.uk

## ABSTRACT

**Objectives:** The optimal method of identifying people with chronic obstructive pulmonary disease (COPD) from electronic primary care records is not known. We assessed the accuracy of different approaches using the Clinical Practice Research Datalink, a UK electronic health record database.

**Setting:** 951 participants registered with a CPRD practice in the UK between 1 January 2004 and 31 December 2012. Individuals were selected for ≥1 of 8 algorithms to identify people with COPD. General practitioners were sent a brief questionnaire and additional evidence to support a COPD diagnosis was requested. All information received was reviewed independently by two respiratory physicians whose opinion was taken as the gold standard.

**Primary outcome measure:** The primary measure of accuracy was the positive predictive value (PPV), the proportion of people identified by each algorithm for whom COPD was confirmed.

**Results:** 951 questionnaires were sent and 738 (78%) returned. After quality control, 696 (73.2%) patients were included in the final analysis. All four algorithms including a specific COPD diagnostic code performed well. Using a diagnostic code alone, the PPV was 86.5% (77.5–92.3%) while requiring a diagnosis plus spirometry plus specific medication; the PPV was slightly higher at 89.4% (80.7–94.5%) but reduced case numbers by 10%. Algorithms without specific diagnostic codes had low PPVs (range 12.2–44.4%).

**Conclusions:** Patients with COPD can be accurately identified from UK primary care records using specific diagnostic codes. Requiring spirometry or COPD medications only marginally improved accuracy. The high accuracy applies since the introduction of an incentivised disease register for COPD as part of Quality and Outcomes Framework in 2004.

## INTRODUCTION

Chronic obstructive pulmonary disease (COPD) represents an enormous health burden worldwide. Currently, COPD is the fourth leading cause of death and is predicted to become the third by 2020.[1] There are approximately 835 000 people diagnosed with COPD in the UK and an estimated 2 200 000 people remain undiagnosed.[2 3]

Electronic health records in the UK provide an excellent resource in which to study COPD as they offer a large cohort size, the presence of disease severity indicators and long-term follow-up information on a patient's integrated medical history. Although multiple studies have been undertaken to investigate various aspects of COPD over the past 10 years in several electronic health record databases, there is no standard definition used to identify COPD in large databases and codelists used to identify patients with COPD vary by author. Over 10 years ago, the diagnosis of COPD was validated in the Clinical Practice Research Datalink-Global initiative for Chronic

### Strengths and limitations of this study

- We have shown that the presence of a specific chronic obstructive pulmonary disease (COPD) Read code alone is sufficient to identify patients with COPD from electronic health records. Minimal precision lost by not including spirometry and medications in the algorithm allows an increase in the number of individuals who can potentially be included in a study by up to 10%.
- We were able to investigate the accuracy of algorithms when identifying patients with COPD within the CPRD, and the accuracy of the actual general practitioner diagnosis of COPD.
- The amount of missing data among the responding questionnaires was low, suggesting reasonable data quality.
- Although the overall response rate for this study was acceptable (77.6%), the proportion of questionnaires accompanied by additional evidence allowing for adjudication was lower.

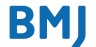

Obstructive Lung Disease (CPRD-GOLD, formerly GPRD) using OXMIS codes.[4] This coding system is now obsolete in CPRD and Read codes are used as the standard.

There is no single diagnostic test for COPD. The diagnosis of COPD relies on clinical judgement based on a combination of history, physical examination and confirmation of the presence of airflow obstruction using spirometry.[5] When retrospectively evaluating the accuracy of a COPD diagnosis, commonly used COPD definitions may misclassify patients as having COPD. Using multiple diagnostic codes in combination with pharmacy may improve the accuracy of identification of patients with COPD.[6] Further, over the past 10 years, the definition of COPD in clinical practice has evolved, leading to changes in how COPD is diagnosed and the diagnosis recorded. With the introduction of the Quality and Outcomes Framework (QOF) codes for COPD in 2004 in England and Wales, there are more codes available to identify COPD then there were previously. QOF is a voluntary incentive scheme for primary care physicians (general practitioners, GPs) in the UK which contains indicators against which GP practices can score points and hence be rewarded for how well they care for patients.[7] Evaluation of airflow limitation using spirometry is used as the standard to confirm COPD diagnosis and severity of COPD is part of the patient with annual COPD review.[3 8] However, even spirometry, if incorrectly performed or interpreted, can lead to misdiagnosis (both overdiagnosis and underdiagnosis of COPD) in approximately 20% of cases. It is also recognised that some subgroups of individuals (eg, women and individuals under 50 and over 80) are less likely to have spirometry measured.[8]

This study aimed to improve our ability to identify patients with COPD within electronic health records. We used the CPRD, a widely used collection of computerised medical records, which is commonly used for clinical and research purposes. CPRD is comparable to other electronic health record databases used in the UK. Our objective was to test the accuracy of different definitions of COPD in the CPRD using the positive predictive value (PPV), by comparing the database records with additional information provided by GPs. This work is important for epidemiological research in COPD and diseases where COPD is an important comorbidity as well as for clinical practice.

## METHODS
### Data set
CPRD is the world's largest validated computerised database of anonymised longitudinal medical records for primary care.[9 10] Data comprise approximately 14 million patients with around 5.4 million of these being currently alive and registered from 660 primary care practices spread throughout the UK. Records are derived from a widely used GP software system and contain complete prescribing and coded diagnostic and clinical information as well as information on tests requested, laboratory results and referrals made at or following on from each consultation.[11]

### Codelists and algorithms
Lists of medical codes (Read codes) specific and non-specific for COPD were created prior to the study initiation. Read codes are a hierarchical clinical coding system of over 80 000 terms that are used in general practice in the UK and are entered by the GP into Vision Software.[12] These data are then uploaded by CPRD after they have been processed, quality checked and added to the CPRD database for research use. Specific COPD codes consisted of codes listing either COPD or COPD-specific entities of emphysema (see online supplementary material for specific codes included). Non-specific codes consisted of a variety of lung diseases that could potentially be COPD, for example, chronic bronchitis. Combinations of codelists and additional material in the form of the presence of spirometry or COPD medications (see online supplementary material) were used to make up the eight algorithms. The first four definitions required a specific COPD diagnosis code, with the first three requiring additional documentation (eg, medication and/or spirometry). The other four definitions required non-specific bronchitis or respiratory symptom codes, with the least specific definition requiring only respiratory symptom codes. Details regarding each algorithm can be found in the online supplementary material. Briefly, the eight algorithms were defined as follows, from the expected most specific to most sensitive construct:

1. Specific COPD code and more than one prescription of a COPD medication and presence of spirometry (*COPD Code+spirometry+COPD medication*);
2. Specific COPD code and presence of spirometry (*COPD Code+spirometry*);
3. Specific COPD code and more than one prescription of a COPD medication (*COPD Code+COPD medication*)
4. Specific COPD code only (*COPD Code only*);
5. Non-specific bronchitis code and more than one prescription of a COPD medication (*Bronchitis+COPD medication*);
6. Non-specific bronchitis code only (*Bronchitis only*);
7. Respiratory symptoms and presence of spirometry. Respiratory symptoms consisted of persistent cough, sputum production or dyspnoea (*Symptoms +spirometry*);
8. Respiratory symptom definition only (*Symptoms only*).

The presence of spirometry for inclusion in the algorithm was based on the existence of a record of a specific value or a Read code for spirometry rather than an examination of the actual value. We were able to assess the interpretation of spirometry in the subset of patients who had flow volume loops or clinic letters attached and where the actual forced expiratory volume in 1 s and forced vital capacity values were available in CPRD.

## Study population

The study population consisted of a random sample of individuals selected from all participants registered in CPRD with the cohort entry being on or after 1 January 2004. At cohort entry, people included had to be: over 35 years old, with at least 1 year historical data, and a possible diagnosis of COPD defined as evidence of having ever smoked and a record of at least one specific or possible COPD code or respiratory symptoms suggestive of COPD. The presence of the algorithms was ascertained during a period between patient cohort entry and 31 December 2012. Patients had to be alive within 4 months of the last collection date of CPRD data for the January 2013 data build for inclusion in the analysis so that CPRD could access their medical records and additional information. For the main analysis, a patient could contribute to one algorithm only. It was possible for an individual to be eligible for more than one algorithm depending on the codes used in their medical record over the study period. Individuals were randomly selected from the algorithm with the fewest number of participants first and then removed from the cohort so that they could not be selected for another algorithm.

CPRD mailed a short, structured questionnaire to GPs in charge of randomly selected patients requesting confirmation of COPD status as well as any available specific information from the individual's medical record including spirometry printouts and hospital respiratory outpatient letters (see online supplementary material). Data were 'twice encrypted' within CPRD to ensure anonymity, first between practices and CPRD and second from CPRD to researchers. In the questionnaire, the GP was asked whether or not the patient had a diagnosis of COPD, what that diagnosis was based on, whether or not the patient had seen a respiratory physician and if they had, whether there were any other respiratory diagnoses. A pilot set of 20 questionnaires were sent to GPs to assess the quality of the questionnaire. In total, 951 questionnaires were sent out, assuming an 80% response rate.

## Primary outcome

The primary outcome was identification of a diagnosis of COPD according to the predefined eight algorithms. The gold standard for the diagnosis of COPD was the decision made after respiratory physicians independently reviewed the evidence from the GP (questionnaire response with or without additional evidence). Where they did not agree, a third independent physician decided. Additionally, GP diagnosis of COPD was validated in a subset of patients where the GP provided supportive information including spirometry printouts and hospital letters. This also allowed a review of spirometry interpretation in some cases. Although we used two respiratory physicians independently as the gold standard for diagnosing COPD, this was done by reviewing the questionnaire sent to the GP (see appendix) as well as any additional information the GP sent which supported that diagnosis. This supporting information ranged from free text in the GP database to spirometry printouts done in the GP surgery to letters from secondary care. Therefore, we were not solely relying on information from secondary care to make the diagnosis unless the GP decided to share that information. In this way, we were able to include and validate the diagnosis of COPD in people who were not seen in secondary care.

## Analysis

The primary analysis focused on the accuracy of identification of a COPD diagnosis in each of the predefined algorithms as defined by the PPV, that is, the proportion of 'true positives' (individuals with COPD) in each algorithm as determined by the gold standard. In addition, within each algorithm, where additional information was provided (lung function, hospital clinic letters), we calculated the accuracy of the GP diagnosis of COPD relative to the gold standard. This allowed a review of spirometry interpretation in some cases.

We assessed the impact of commonly occurring comorbidities on the accuracy of the prespecified algorithms stratifying for cardiovascular comorbidity, previous asthma diagnosis, smoking status and, where possible, GOLD staging of airflow limitation severity.[13] Cardiovascular comorbidity included angina, history of myocardial infarction, previous coronary artery bypass graft/percutaneous coronary intervention or heart failure, but not hypertension due to its lack of overlap of symptoms that could mimic COPD. All covariates for stratification analysis were derived from information available up to cohort entry.

As a post hoc analysis, individuals were eligible to be placed into multiple algorithms where possible, and the PPV was calculated for all individuals who had a specific COPD code compared with those with a specific COPD code and additional information (either spirometry or a COPD medication).

Assessment of possible trends in COPD diagnosis recording were also evaluated, including temporal trends in codes used and diagnostic specificity from 2004 to 2011. In addition, we compared our specific COPD codes with those recommended for use by QOF (see online supplementary material)[14 15]: H31% (excluding H3101 (smoker's cough), H31y0 (chronic tracheitis) and H3122 (acute exacerbation of COPD)) and H32% H36-H3z (excluding H3y0 and H3y1).

## Sample size calculation

Our sample size for each algorithm was chosen to achieve accuracy of the true positives or the PPV ±0.08 based on the reviewing physician's judgement as the gold standard. Assuming an estimated PPV of 0.85 for any one algorithm, we required a sample of at least 77 individuals in each algorithm to achieve the desired accuracy (95% CI ±0.08). All analyses were performed using STATA V.13.

## RESULTS

Nine hundred and fifty-one questionnaires were sent to GPs (see figure 1 for patient selection). Of those, 738 (77.6%) were returned, 704 (74.0%) met quality control standards and were not duplicates and 696 (73.2%) could be included in the final analysis (8 had 'uncertain' COPD diagnosis and no supporting documentation and were therefore excluded).

Among those included in the final analysis, additional evidence for the diagnosis of COPD was available for 272 patients. This represented 39.1% of the total study population, or 67.7% of the 402 patients with a confirmed COPD diagnosis in the study.

Overall, irrespective of the qualifying algorithm, 402 patients (57.8%) were considered to have a diagnosis of COPD based on the reviewing physician judgement. Table 1 shows the characteristics of the 696 patients included in the final analysis who were considered to possibly have COPD based on the inclusion criteria. On average, patients were in their mid-60s to early 70s across all algorithms. Approximately two-thirds of them were current smokers and one-quarter had a history of asthma. Generally, there were fewer patients with

supporting information and cardiovascular comorbidity in the less specific algorithms (4–8).

The number of patients diagnosed with COPD confirmed by the gold standard and the PPV for each algorithm is given in table 2. The PPV was greatest for algorithms 1–4. Further data are available in the online supplementary material regarding the effect of comorbidities (see online supplementary table S1), smoking status (see online supplementary table S2) and GOLD staging (see online supplementary table S3) on the performance of each algorithm.

In a subset of 272 patients where additional evidence was available (in the form of spirometry printouts or hospital outpatient letters), we assessed accuracy of GP diagnosis of COPD. Overall, the PPV in this group was 95% (91.1–97.2). This is broken down by the algorithm in table 3. While the presence of supporting evidence improved the PPV in each group, algorithms 1–4 were still most accurate.

### Post hoc analysis

We repeated the analysis allowing individuals to be put into more than one algorithm and tested the PPV of

**Figure 1** Study population (COPD, chronic obstructive pulmonary disease; GP, general practitioner).

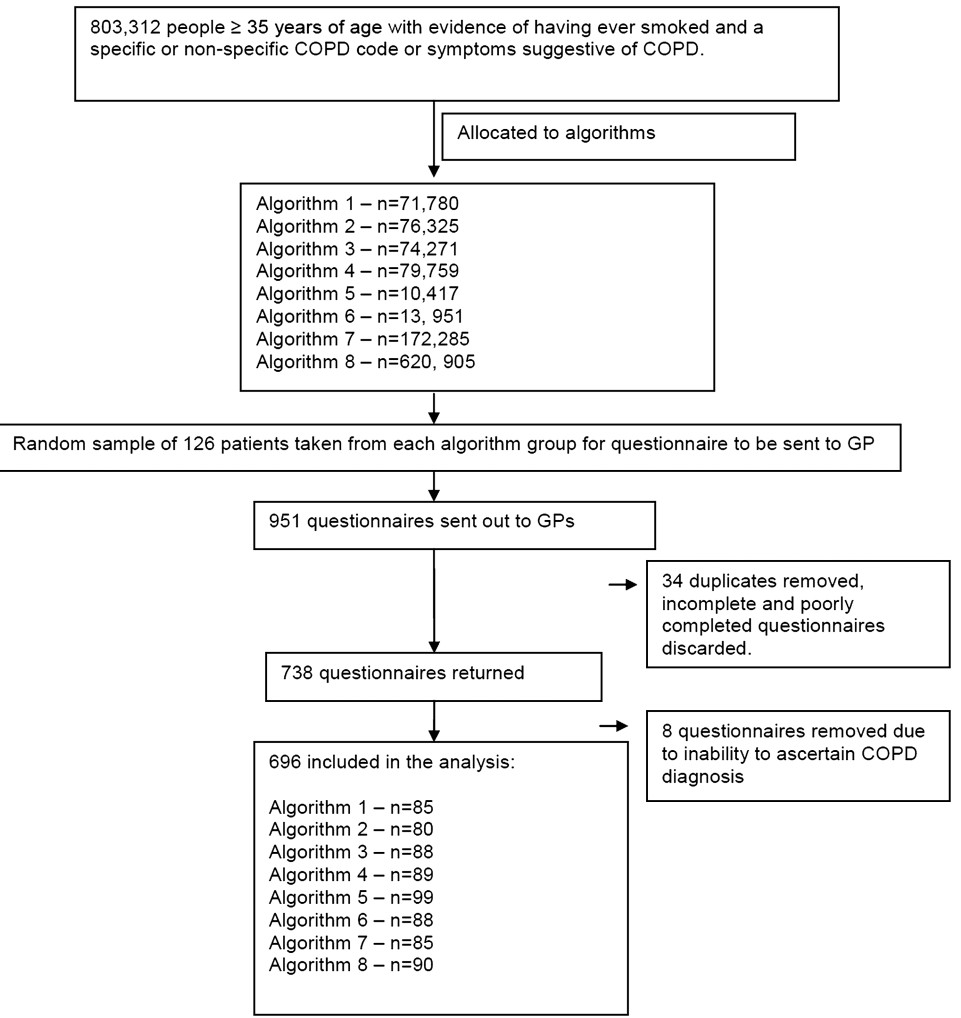

**Table 1** Characteristics of the 696 patients included in the final study analysis

| Algorithm | COPD Code +spirometry+COPD medication | COPD Code +spirometry | COPD Code +COPD medication | COPD Code only | Bronchitis +COPD medication | Bronchitis only | Symptoms +spirometry | Symptoms only |
|---|---|---|---|---|---|---|---|---|
| Number of individuals (%) | 85 (100) | 79 (100) | 88 (100) | 89 (100) | 98 (100) | 84 (100) | 83 (100) | 90 (100) |
| Number (%) with supporting info | 46 (54.1) | 44 (55.7) | 48 (54.5) | 40 (44.9) | 32 (32.7) | 18 (21.4) | 30 (36.1) | 14 (15.6) |
| Mean age (SD) | 68.7 (11.3) | 68.3 (11.7) | 71.8 (10.5) | 71.1 (10.4) | 68.5 (13.1) | 67.8 (13.4) | 65.9 (11.9) | 63.4 (14.1) |
| Male (%) | 45 (52.9) | 41 (51.9) | 40 (45.5) | 44 (49.4) | 31 (31.6) | 29 (34.5) | 43 (51.8) | 47 (52.2) |
| Current smoker (%) | 49 (57.7) | 50 (63.3) | 55 (62.5) | 47 (52.8) | 66 (67.4) | 61 (72.6) | 48 (57.8) | 58 (64.4) |
| GOLD stage (n=465)* | | | | | | | | |
| 1 | 13 (16.3) | 14 (18.0) | 13 (15.9) | 16 (20.5) | 17 (28.8) | 13 (35.1) | 8 (20.0) | 4 (36.4) |
| 2 | 43 (53.8) | 48 (61.5) | 41 (50.0) | 46 (59.0) | 31 (52.5) | 13 (35.1) | 22 (55.0) | 5 (45.5) |
| 3 | 18 (22.5) | 13 (16.7) | 22 (26.8) | 12 (15.4) | 9 (15.3) | 8 (21.6) | 8 (20.0) | 2 (18.2) |
| 4 | 6 (7.5) | 3 (3.9) | 6 (7.3) | 4 (5.1) | 2 (3.4) | 3 (8.1) | 2 (5.0) | 0 (0.0) |
| History of cardiovascular disease | 20 (23.5) | 22 (27.8) | 17 (19.3) | 18 (20.2) | 20 (20.4) | 17 (20.2) | 14 (16.9) | 9 (10.0) |
| History of asthma | 18 (21.2) | 20 (25.3) | 16 (18.2) | 15 (16.9) | 27 (27.6) | 23 (27.4) | 19 (22.9) | 23 (25.6) |
| Mean BMI (SD) (n=575) | N=83 | N=78 | N=86 | N=87 | N=98 | N=85 | N=41 | N=17 |
| | 27.5 (5.4) | 26.7 (5.8) | 26.4 (4.7) | 27.8 (5.4) | 27.4 (5.1) | 28.1 (5.0) | 27.1 (4.5) | 27.6 (4.7) |

*GOLD staging was ascertained from CPRD records or from supplementary information provided by GPs.
BMI, body mass index; COPD, chronic obstructive pulmonary disease; CPRD, Clinical Practice Research Datalink; GOLD, Global initiative for Chronic Obstructive Lung Disease; GP, general practitioner.

**Table 2** The positive predictive value (PPV) and proportion of patients diagnosed with chronic obstructive pulmonary disease (COPD) within each algorithm

| Algorithm | Number of questionnaires sent out (n=951) | Number evaluable returned (n=696) (%) | Number with confirmed COPD | PPV and 95% CI |
|---|---|---|---|---|
| COPD Code+spirometry +COPD medication | 119 | 85 (71.4) | 76 | 89.4, 80.7 to 94.5 |
| COPD Code+spirometry | 119 | 79 (66.4) | 67 | 83.8, 73.7 to 90.4 |
| COPD Code+COPD medication | 119 | 88 (73.9) | 77 | 87.5, 78.6 to 93.0 |
| COPD Code only | 119 | 89 (74.8) | 77 | 86.5, 77.5 to 92.3 |
| Bronchitis+COPD medication | 119 | 98 (82.4) | 44 | 44.4, 34.8 to 54.5 |
| Bronchitis only | 119 | 84 (70.6) | 26 | 29.5, 20.8 to 40.1 |
| Symptoms+spirometry | 119 | 83 (69.7) | 37 | 43.5, 33.2 to 54.4 |
| Symptoms only | 118 | 90 (75.6) | 11 | 12.2, 6.8 to 20.9 |

COPD by the algorithm relative to the gold standard where individuals were identified using only the presence of a specific COPD code (n=457) the PPV was identical to if they had a COPD code and evidence either in the form of spirometry or COPD medication prescription (n=454); PPV 83% (79–86) for both. In the majority of cases, where a specific COPD code had been assigned, there was additional evidence. Only three individuals had a specific COPD code with no additional evidence.

## DISCUSSION

We tested the accuracy of eight different algorithms for identifying COPD within the CPRD among patients with suspected COPD (eg, >35 years of age, smoking history and recording of respiratory symptoms or COPD codes). The physician reviewer's consensus was the gold standard. The best performing algorithm allowed an accurate ascertainment of 90% of patients as diagnosed with COPD. This consisted of a combination of a specific COPD code, more than one prescription of a COPD medication and spirometry (PPV 89.4, 95% CI 80.7 to 94.5). The worst performing algorithm was represented by the presence of respiratory symptoms only (PPV 12.2, 95%CI 6.8 to 20.9). We found that any algorithm containing a specific COPD code performed better than those without (algorithms 1–4). In a post hoc analysis, where we allowed individuals to populate more than one algorithm, we established that the use of additional information such as spirometry or medications in an algorithm to a specific COPD Read code alone did not increase the PPV. This suggests that the presence of a specific COPD Read code alone is sufficient to accurately identify patients with COPD from the database. Some study protocols require the presence of obstructive spirometry for identification of patients with COPD; however, this study demonstrates that it may be unnecessary. This is particularly important as certain groups of individuals are less likely to have spirometry, namely

**Table 3** Positive predictive value (PPV) by algorithm where evidence was available to assess GP compared with physician diagnosis of chronic obstructive pulmonary disease (COPD)

| Algorithm | Total number with evidence (N=272) | Number of patients with COPD confirmed by gold standard (N=220) | Number with COPD according to GP | PPV and 95% CI |
|---|---|---|---|---|
| COPD Code +spirometry+COPD medication | 46 | 46 | 46 | 100 |
| COPD Code +spirometry | 44 | 35 | 33 | 94.3, 82.4 to 98.9 |
| COPD Code+COPD medication | 48 | 43 | 41 | 95.3, 82.4 to 98.9 |
| COPD Code only | 40 | 34 | 33 | 97.1, 80.2 to 99.6 |
| Bronchitis+COPD medication | 32 | 21 | 19 | 90.5, 66.0 to 97.9 |
| Bronchitis only | 18 | 12 | 11 | 91.7, 49.9 to 99.2 |
| Symptoms+spirometry | 30 | 21 | 20 | 95.2, 69.1 to 99.4 |
| Symptoms only | 14 | 8 | 6 | 75.0, 27.6 to 95.9 |

women and patients <50 and >80 years of age.[8] This is also important as the minimal precision lost by not including spirometry and medications in the algorithm allows an increase in the number of individuals who can potentially be included in a study. Using the whole of CPRD, we identified individuals with COPD using a specific COPD code only compared with a specific code plus medication and spirometry and found an increase in the potential sample size for a study of 10% using a specific COPD code only.

One of the advantages of this study was our ability to investigate the accuracy of algorithms when identifying patients with COPD within the CPRD, and the accuracy of the actual GP diagnosis of COPD. When validating the GP diagnosis of COPD with a respiratory physician's diagnosis as the gold standard based on extra evidence provided by the GP, there was improved accuracy (PPV) across all algorithms, with algorithms 1–4 again performing best. This suggests that additional evidence is collected when the GP is reasonably certain that the patient has COPD. There was good concordance between the GP's and the respiratory physician's diagnosis, suggesting that respiratory consultant validation is not always needed. Where there was disagreement, this was usually because lung function did not meet the criteria for COPD.

We found that the diagnostic accuracy of COPD decreased across all algorithms when patients also had a diagnosis of cardiovascular disease or asthma (see online supplementary table S1). When patients had a concomitant diagnosis of asthma, the presence of spirometry was particularly important to improve accuracy of COPD diagnosis. This was predominantly due to the fact that spirometry had been misinterpreted. However, stratification led to smaller sample sizes, which could have impacted the precision of estimates. Unsurprisingly, the addition of the use of any inhaled COPD medication to the algorithm did not improve precision, most likely due to the overlap in medications used to treat asthma and COPD (see online supplementary material). The algorithm accuracy was not affected by smoking status (current vs ex-smoker; see online supplementary table S2). We only included current or former smokers in our analysis, and cannot be sure of the validity of the results in a patient who has never smoked. Certainly in the UK, the majority of COPD is related to tobacco smoking and we hypothesised that fixed airflow obstruction in a non-smoker would most likely be due to chronic asthma.

When considering the severity of COPD by GOLD classification, algorithms 2 and 3, that is, a specific COPD code and spirometry or COPD medications, had the greatest accuracy for patients with mild disease (GOLD stage 1; see online supplementary table S3). The PPV increased with increasing disease severity. It increases with disease prevalence, and the prevalence of COPD increased by moving from algorithm 8 to algorithm 1.

We considered the timing of diagnosis in view of an increased uptake of spirometry in primary care in more recent years and changes in QOF requirements over time during our study period (see online supplementary table S4). We found that algorithms 1–4 still had the greatest accuracy, but the PPV estimates were better for the post 2008 period than the pre 2008 period (see online supplementary material). Non-specific bronchitis codes and symptom codes were more likely to be used before 2008 rather than after 2008. This may require consideration when developing retrospective cohorts for analysis and otherwise suggests that QOF has had a positive effect on the consistency of codes used for COPD diagnoses. However, our codes were more specific than QOF codes and some of the codes included in QOF were not included in our specific COPD codelist, but were included in the non-specific bronchitis codes. While we cannot comment on the accuracy of QOF codes, it is important to highlight that some QOF codes are not disease specific and may not be a good way of identifying patients with COPD from electronic health records as inevitably people without COPD will be included. It is also important to highlight that Read codes change over time with new codes being added and some removed, and this needs consideration when identifying people with COPD.

There are limited COPD validation studies in electronic health records published in the literature with which we can compare our study. Soriano et al[4] validated COPD in the GPRD in 2001, when OXMIS codes were still in use. A Swedish study using a Swedish inpatient registry used International Classification of Diseases (ICD)-9 and ICD-10 codes and identified patients with COPD with similar accuracy.[16] A Canadian study in the Canadian primary care sentinel surveillance network used algorithms to identify several long-term conditions and also had a PPV similar to ours for COPD.[17] However, all of these studies used different codes, algorithms and databases.

Our analysis has several limitations. We appreciate that using the gold standard of a GP questionnaire and respiratory physician review is not perfect as when asked about whether or not a specific patient has a certain diagnosis, the GP is most likely to simply look in the electronic health record and see if that diagnosis has been recorded. However, there is no alternative. The overall response rate for this study was acceptable (77.6%), while the proportion of questionnaires accompanied by additional evidence allowing for adjudication was rather low. We used PPV in this study as the measure of accuracy to allow us to determine the probability that a patient had COPD from their electronic health record. The PPV is correlated with disease prevalence, and although it is strongly related to specificity, the actual estimates of specificity, sensitivity and negative predictive value cannot be determined from our data. Further, GP practices are self-selecting with respect to their contribution to CPRD; however, those practices appear to be representative of the UK population. Very few patients within contributing practices refuse to participate at an

individual level and this is not thought to bias the results. While CPRD is representative of the general population, as with all validation studies that require a response, we cannot be sure that our sample is representative of GPs who have not responded, although there is unlikely to be any difference. The amount of missing data among the responding questionnaires was low, suggesting reasonable data quality. One of the other limitations of this study is that patients had to be alive to be included; however, it is unlikely that coding would be different for individuals who are no longer alive.

The algorithm that consisted of a specific COPD code, COPD medication and spirometry had the highest PPV; however, the PPV was almost as high when a specific COPD code alone was used. The poorest performing algorithms were those that involved bronchitis codes or respiratory symptoms; we would not recommend using these algorithms to identify patients with COPD. In conclusion, we have shown that the presence of a specific COPD Read code alone is sufficient to identify patients with COPD from electronic health records such as CPRD. Minimal precision lost by not including spirometry and medications in the algorithm allows an increase in the number of individuals who can potentially be included in a study by up to 10%. However, by not including spirometry in the definition, the ability to stage COPD according to GOLD stages may not be possible for all patients with COPD included in a study.

**Author affiliations**
[1]Department of Non-Communicable Disease Epidemiology, London School of Hygiene and Tropical Medicine, London, UK
[2]Department of Respiratory Epidemiology, GlaxoSmithKline R&D, Uxbridge, UK
[3]Department of Respiratory Epidemiology, GlaxoSmithKline R&D, ResearchTriangle Park, North Carolina, USA
[4]Clinical Practice Research Datalink Group, Medicines and Healthcare Products Regulatory Agency, London, UK
[5]Department of UCL Respiratory Medicine, Royal Free Campus, University College London Medical School, London, UK

**Contributors** JKQ, HM, RLD, SE, KD and LS contributed to the conception and design. JKQ, HM and JH contributed to the acquisition and analysis. JKQ, HM, RLD, HF, SE, JRH, KD and LS contributed to the interpretation of data. JKQ, HM, RLD, HF, SE, JRH, KD and LS contributed to the drafting of the manuscript. JKQ is responsible for the overall content as guarantor.

**Funding** This project was funded jointly by GSK and the MRC. This work was funded by an MRC Industry Partnership award (grant number G0902135). JKQ is funded on a MRC Population Health Scientist Fellowship.

**Competing interests** None.

**Ethics approval** Ethics approval was obtained from ISAC (the Independent Scientific Advisory Committee overseeing CPRD), protocol 12_065 and the LSHTM ethics committee.

**Provenance and peer review** Not commissioned; externally peer reviewed.

**Data sharing statement** No additional data are available.

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
