## [Reviewer comments · BMJ Open]

This paper was submitted to the Thorax but declined for publication following peer review. The authors addressed the reviewers' comments and submitted the revised paper to BMJ Open. The paper was subsequently accepted for publication at BMJ Open.

ARTICLE DETAILS

TITLE (PROVISIONAL)	Validation of Chronic Obstructive Pulmonary Disease (COPD) recording in the Clinical Practice Research Datalink (CPRD-GOLD)
AUTHORS	Quint, Jennifer; Mullerova, Hana; DiSantostefano, Rachael; Forbes, Harriet; Eaton, Susan; Hurst, John; Davis, Kourtney; Smeeth, Liam

VERSION 1 - REVIEW

REVIEWER	van der Molen, Thys University Medical Center Groningen
REVIEW RETURNED	31-Mar-2014

GENERAL COMMENTS	General comments: The article is well written but very difficult to understand when one is not aware of background information regarding QOF and data from General practice. Second comment: The gold standard however important is not evaluated. Clinicians often differ in opinion about diagnosis my suggestion would be to add a paragraph with data about the gold standard. Abstract: difficult to understand for a foreign GP Introduction: Paragraph 1 line 4 Reference 2: I might have the wrong Shahab but in pubmed this is Shahab L and not LM Nevertheless the calculation of 2.200.000 undiagnosed COPD is not from this article but from calculations of the Authors? Furthermore Shahab at all in their discussion admit that this is including Asthma and the spirometry is made without bronchodilation. In my opinion these numbers are not valid please check or comment. Codelists and algorithms section, For non British readers it might be extremely difficult to understand this. What is the QOF what happens in the GP practice how is this coded? These kind of questions are yet unanswered in the article. Without this information the clinician from abroad will not understand the section. What is a COPD code ? Is this what the GP states when he or she makes the diagnosis COPD? What are the so called COPD medications? I see this is answered in the additional information. Why are low dose ICS included? These codes are also applicable to asthma.? Does the GP get additional payment for a patient with COPD by using QOF
--

	Is the spirometry with or without reversibility? How was the gold standard evaluated? What kind of information did the two pulmonologists get and in how many cases did they differ in opinion and why the third clinician? Why not a next patient. Can patients be excluded when there is difference in opinion? Does this change the conclusions of the article? Results 25 % of patients had asthma do these patients count as COPD patients? Discussion, The first part of the discussion states that GPs when giving the COPD code- thus actually giving the diagnosis of COPD- this is the best predictor of a diagnosis of COPD given by the pulmonologist. This is no wonder since the pulmonologists in this case depend on the information given by the GP. Could the authors please comment?
--	---

VERSION 1 – AUTHOR RESPONSE

General comments: The article is well written but very difficult to understand when one is not aware of background information regarding QOF and data from General practice. **We have added further background information in the introduction to clarify this further.**

“QOF is a voluntary incentive scheme for primary care physicians (GPs) in the UK which contains indicators against which GP practices can score points and hence be rewarded for how well they care for patients.”

Second comment: The gold standard however important is not evaluated. Clinicians often differ in opinion about diagnosis my suggestion would be to add a paragraph with data about the gold standard.

We have further clarified the gold standard used.

“Although we used 2 respiratory physicians independently as the gold standard for diagnosing COPD, this was done by reviewing the questionnaire sent to the GP (see appendix) as well as any additional information the GP sent which supported that diagnosis. This supporting information ranged from free text in the GP database to spirometry printouts done in the GP surgery to letters from secondary care. Therefore we were not solely relying on information from secondary care to make the diagnosis unless the GP decided to share that information. In this way we were able to include and validate the diagnosis of COPD in people who were not seen in secondary care. “

Abstract: difficult to understand for a foreign GP

We have reworded the abstract.

Introduction:

Paragraph 1 line 4 Reference 2: I might have the wrong Shahab but in pubmed this is Shahab L and not LM Nevertheless the calculation of 2.200.000 undiagnosed COPD is not from this article but from calculations of the Authors? Furthermore Shahab at all in their discussion admit that this is including Asthma and the spirometry is made without bronchodilation. In my opinion these numbers are not valid please check or comment. **We have added a further reference to clarify this. This sentence**

refers to the “missing millions” with undiagnosed COPD in the UK.

Codelists and algorithms section, For non British readers it might be extremely difficult to understand this. What is the QOF what happens in the GP practice how is this coded? These kind of questions are yet unanswered in the article. Without this information the clinician from abroad will not understand the section. What is a COPD code ? Is this what the GP states when he or she makes the diagnosis COPD?

We have added further information on QOF to the paper in the introduction. We have added more information on Read codes to the codelist section.

“Read codes are a hierarchical clinical coding system of over 80,000 terms that are used in general practice in the UK and are entered by the GP into Vision Software. These data are then uploaded by CPRD after they have been processed and quality checked added to the CPRD database for research use.”

What are the so called COPD medications? I see this is answered in the additional information. Why are low dose ICS included? These codes are also applicable to asthma.?

The COPD medications are in the additional information section. The list was too long to include in the main paper. Some of the medications are also applicable to asthma. We included low dose ICS as we did not always have dosing information available and because we know that although they should not be used for COPD they sometimes are.

Does the GP get additional payment for a patient with COPD by using QOF

Yes

Is the spirometry with or without reversibility?

According to QOF guidance, spirometry should be post bronchodilator and reversibility is only done if asthma is suspected. We have therefore assumed this to be the case.

How was the gold standard evaluated? What kind of information did the two pulmonologists get and in how many cases did they differ in opinion and why the third clinician? Why not a next patient. Can patients be excluded when there is difference in opinion? Does this change the conclusions of the article?

Although we used 2 respiratory physicians independently as the gold standard for diagnosing COPD, this was done by reviewing the questionnaire sent to the GP (see appendix) as well as any additional information the GP sent which supported that diagnosis. This supporting information ranged from free text in the GP database to spirometry printouts done in the GP surgery to letters from secondary care. Therefore we were not solely relying on information from secondary care to make the diagnosis unless the GP decided to share that information. In this way we were able to include and validate the diagnosis of COPD in people who were not seen in secondary care. This has been clarified in the manuscript. There was only disagreement on 4 cases between both respiratory physicians regarding a COPD diagnosis and in the end after discussion with the 3rd physician these 4 cases were removed from the analysis as it was not possible to ascertain if those patients had a diagnosis of COPD or not. The conclusions of the article were not changed by removing those 4 patients.

Results

25 % of patients had asthma do these patients count as COPD patients? Yes. Irrespective of a diagnosis of asthma, if the GP said the patient had COPD then the questionnaire and supporting information were assessed and a conclusion made as to whether or not they had COPD. All of the patients had a significant smoking history. We know that some patients do have both diagnoses and we appreciate that the accuracy of the diagnosis was lower in patients who had asthma and COPD and so we specifically looked at this subgroup of patients (see supplementary material for table).

Discussion,

The first part of the discussion states that GPs when giving the COPD code- thus actually giving the

diagnosis of COPD- this is the best predictor of a diagnosis of COPD given by the pulmonologist. This is no wonder since the pulmonologists in this case depend on the information given by the GP. Could the authors please comment?

Yes, we were trying to determine how accurate a GP diagnosis of COPD was when using electronic health records to identify patients. We found that only certain Read codes for COPD had good diagnostic accuracy and that minimal precision lost by not including spirometry and medications in the algorithm, allows an increase in the number of individuals who can potentially be included in a study by up to 10%.